# Study on Austenite Transformation and Growth Evolution of HSLA Steel

**DOI:** 10.3390/ma16093578

**Published:** 2023-05-07

**Authors:** Lu Wang, Shaoyang Wang

**Affiliations:** School of Materials, Sun Yat-sen University, Guangzhou 510275, China; wanglu68@mail.sysu.edu.cn

**Keywords:** high strength low alloy steel, austenite transformation, grain growth, precipitate, mixed grain

## Abstract

HSLA steel is widely used in various applications for its excellent mechanical properties. The evolution of austenite transformation and growth has been systematically studied in HSLA steel Q960 during the heating process. A thermal expansion instrument and optical microscope were adopted to analyze the kinetics of austenite transformation, which is a nonlinear continuous process and was accurately calculated by the lever rule based on the dilatation curve at the holding time within 10 min. The austenite growth behavior at temperatures above A_c3_ was explored using TEM and DSC. The main precipitates in austenite were Nb-rich and Ti-rich (Nb, Ti)(C, N), and the particle size increased and amount decreased with the increase in the heating temperature, which resulted in the rapid growth of austenite. With the increase in holding temperature and time, the growth of austenite progressed through three stages, and a heat treatment diagram was established to describe this evolution.

## 1. Introduction

High-strength low alloy (HSLA) steels with higher strength, excellent low-temperature toughness and weldability are widely used in various applications such as naval ship structures, automobiles, bridges, high-pressure vessels and railways [1,2,3]. For HSLA steels, the grain size of the final microstructure is an important parameter to determine the mechanical properties, which depend on the processing conditions of the previous austenitic structure [4]. In industrial enterprises, the continuous casting slab of HSLA steel must undergo proper heat treatment before rolling to fully homogenize the composition and structure and obtain an ideal performance. Additionally, the existence of an iron allotrope transformation in steel enables it to obtain various microstructures and mechanical properties through heat treatment. The austenitizing process of steel materials during heat treatment is also the key to other types of transformation, which is meaningful for the subsequent thermal deformation or heat treatment quality. Lu et al. [5] studied the effect of reverse austenite on the mechanical properties of HSLA steel, where austenite forming on ferrite grain boundaries can significantly improve its toughness. Ghorabaei et al. [6] found out that in ultra-low-carbon HSLA steel when the austenite grain size was refined from ~69 μm to ~10 μm, there existed an isothermal-to-nonisothermal transition, where austenite transformed into various ferritic products of quasi-polygonal ferrite, granular bainite, bainitic ferrite, and martensite. It has been well recognized that the evolution of prior austenite is controlled by a number of factors, e.g., the heating temperature and holding time, dislocation density, texture, growth and dissolution of the precipitates [2,7]. The heating temperature and holding time directly affect the morphology, size, distribution and alloy element distribution of transformed austenite transformation. Chamanfar et al. [8] analyzed the austenite grain growth behavior in low alloy medium carbon forged steel and indicated that the soaking temperature and time were primary controlling variables during austenitization, which affected the final microstructure and mechanical properties. At a high heating temperature, the alloying elements were distributed more uniformly, but the grains grew fast and easily appeared as mixed grains, which made the control of austenite refinement and transformation more difficult, and surface cracks along the austenite grain boundaries and other defects during the hot deformation and cooling process were easily formed on slabs [4,9,10,11]. When the heating temperature was low, the formed austenite grains were even and fine, which helped to reduce the internal stress and refine the structure in the subsequent deformation and transformation process, but the alloy elements could not be fully dissolved, which easily caused structural and composition segregation. Moreover, the carbonitrides in HSLA steel affect the austenite grain growth; for example, Ti-rich rectangular carbonitrides have high thermal stability and can be pinned to austenitic grain boundaries, and the dissolution of Nb in austenite can lead to the rapid growth of austenite grains, which slows down ferrite transformation and is beneficial to the formation of bainite [12,13,14]. Nevertheless, the pinning of precipitates in the segregation band of the microstructure can lead to the abnormal grain growth of austenite during the reheating process [15]. Therefore, a reasonable heating system is of great significance when saving energy, reducing defects, and improving processing and serving performance.

In this work, we systematically studied austenite transformation and growth behavior during the heating process based on a continuous casting slab of Q960 steel. The transformation process of austenite at the inter-critical temperature was characterized by the thermal expansion instrument and by combining the microstructure analysis of a mathematical model, which was developed to predicate austenite transformation. The growth behavior of austenite at temperatures above A_c3_ was discussed in detail, and the effect of precipitates on austenite growth was explored based on TEM and DSC experiments. Finally, the relationship between austenite evolution, heating temperature, and the holding time was obtained, which provided valuable guidance for heat treatment resulting in an optimal microstructure and mechanical properties in HSLA steels.

## 2. Materials and Methods

The material used in this work was taken from a quarter thickness of a continuous casting slab provided by Ma Steel (Anhui, China), which is Q960 steel with chemical composition, as shown in Table 1. The phase transformation process of steel during heat treatment was studied with a German Beahr DIL 805A thermal expansion instrument, which has the highest heating temperature of 1450 °C, the fastest cooling rate of 200 °C/s, and measurement accuracy of 0.05 μm/0.05 °C. Additionally, samples with a size of Φ 4 mm × 10 mm were heated at 10 °C/min to 1200 °C held for 5 min and then rapidly cooled at 3000 °C/min to room temperature through liquid nitrogen. The dissolution temperature of precipitates was measured experimentally using the STA 449C differential scanning calorimetry (DSC), and a sample with a size of Φ 5 mm × 1 mm was heated at a constant rate of 10 °C/min to 1300 °C. Additionally, the precipitates in the steel were theoretically calculated with Thermo-Calc based on the TCFE8 database and according to the chemical composition. In order to analyze the microstructure morphology, a muffle furnace was adopted to conduct heat treatment experiments. Samples with a size of 10 mm × 10 mm × 10 mm were heated to 790 °C, 810 °C, 830 °C, 850 °C and 880 °C and held for 10 min, 30 min and 60 min, respectively, and then cooled to room temperature in cold water to ensure that the high-temperature austenite transformed into martensite. Then, the samples were ground and polished after heat treatment and were etched with 4% nital; the microstructure was analyzed under an optical microscope (Carl Zeiss Microscopy GmbH, Jena, Germany). The fraction of each phase in the microstructure was counted by using the phase analysis software of Image Pro-Plus 6.0 software (Media Cybernetics, Inc., Washington, WA, USA) and by analyzing the quantity and distribution of martensite, the evolution process of austenite transformation during heating could be obtained. Subsequently, a higher-temperature heat treatment was carried out to study the austenite growth and coarsening process. Samples were heated to 1100 °C, 1150 °C, 1200 °C, 1250 °C and 1300 °C and held for 0 min, 30 min, 60 min and 120 min, respectively, in the muffle furnace and then quenched immediately in cold water. Additionally, the quenched samples, after grounding and polishing, were hot and etched in a saturated picric acid solution (60 mL deionized water, 2.5 g picric acid and 1 g sodium dodecylbenzene sulfonate) for 4 min at 80 °C to show the original austenite grain boundary [16]. Then, the original austenite grain size before quenching was analyzed with an optical microscope by measuring more than 200 grains for each sample. After that, the quenched samples after metallographic analysis were carried out with carbon spraying treatment on the surface to extract the precipitates from the sample onto a 3 mm × 3 mm carbon film in 5% nital [17]. Additionally, the extracted carbon film was taken with a Φ 3 mm copper mesh, and the precipitates in the samples were analyzed after heat treatment under a JEM-2100 transmission electron microscope (TEM).

## 3. Results and Discussion

### 3.1. Kinetics of Austenite Transformation

The microstructure of the continuous casting billet mainly consisted of coarse ferrite and pearlite, as shown in Figure 1. The average grain size of ferrite was about 150 μm. The pearlite was evenly distributed among the ferrite grains.

According to the different dilatometric behavior of the phases in steel, phase transformation can cause the volume expansion or contraction of the sample, then the temperature of the phase transformation can be measured by the thermal expansion test [18]. The measured dilatometric curve of the sample length during the heating and cooling process is shown in Figure 2. According to the variation in the dilatometric curve caused by phase transformation, the starting and finishing temperatures of the phase transformation were determined by the inflection point of the curve. Additionally, the austenite transformation temperature of A_c1_ and A_c3_ during heating were 730 °C and 866 °C, respectively. Due to the metallographic structure of the sample after cooling, which was determined to be martensite, the martensite transformation temperature of M_s_ and M_f_ were 410 °C and 300 °C, respectively.

From the dilatometric curve between A_c1_ and A_c3_ in Figure 2, it can be seen that with the increase in the heating temperature, the transformation of pearlite and ferrite to austenite was a nonlinear continuous process, where the pearlite first transformed into austenite at a low temperature (such as 730 °C), and then ferrite began to transform into austenite at a higher temperature (such as 745 °C) [19]. The microstructure of the samples after quenching from the austenitizing temperature is shown in Figure 3, which mainly consists of ferrite and martensite. Since the heating temperatures of 790 °C–850 °C were within the critical transformation temperatures A_c1_ and A_c3_, all the formed austenite transformed into martensite after cooling; the shape, quantity and distribution of martensite in the microstructure corresponded to the state of austenite before quenching. So, the martensite here was used to descript the transformed austenite before cooling. It can be seen that the martensite content gradually increased with the increase in the heating temperature and holding time. Holding at 790 °C for 10 min, the ferrite content was high, with the martensite distributed around and within the ferrite grains, and with the increase in holding time, the ferrite content and grain size decreased. With the increase in the heating temperature, such as to 810 °C and 830 °C, the content and grain size of ferrite decreased continuously, and the distribution of ferrite and martensite tended to be uniform with ferrite distribution around the martensite. When the heating temperature rose to 850 °C, which is close to the critical transformation point A_c3_, the quenched microstructure was nearly all martensite with a small amount of equiaxed ferrite with a small size distributed among the martensite. At this temperature, with the increase in the holding time, the microstructure changed little. After quenching from 880 °C, where the temperature was greater than A_c3_ and the initial ferrite and pearlite completely transformed into austenite, the structure was only martensite, and with the increase in the holding time, the change in martensite was not obvious. 

Since the microstructure only contains ferrite and austenite at the holding temperature, austenite content under different heating conditions could be obtained by counting the ferrite fraction in the microstructure of Figure 3, and the results are shown in Figure 4. For the same holding time, with the increase in the heating temperature the amount of austenite increased. When held at 880 °C, the fraction of austenite almost remained at 100%, which also indicates that the critical temperature A_c3_ was lower than 880 °C. At the same temperature, the fraction of the formed austenite increased with prolonged holding time, and the lower the heating temperature, the slower the rate of austenite transformation, resulting in a longer time with which to complete austenite transformation. For example, the sample holding at 790 °C needed more than 60 min to complete the transformation of austenite, while needed about 30 min for the sample holding at 850 °C to finish the austenite transformation. The transformation of ferrite and lamellar pearlite to austenite is a diffusion process of carbon atoms, which can be accelerated by a high temperature; therefore, austenite transforms fast with the increase in the heating temperature. 

According to the dilatometric curve of the austenite transformation process, the lever rule was employed to calculate the amount of austenite formation (*f*_γ_) at the temperature between A_c1_ and A_c3_, as shown in Figure 5a [20].
(1)$$f_{\gamma }=\frac{x}{x+y}$$
where *x* and *y* are the lengths of the vertical segments above and below the dilatometric curve, respectively. Figure 5b shows the relationship between the austenite fraction, heating temperature and time. It can be seen that the calculated value of austenite content was in good agreement with the results measured when held for 10 min at the same temperature but was lower than the results measured after holding for a longer time. Because during the continuous heating process in the thermal expansion instrument, the holding time at each temperature was very short, the calculated results were close to that measured after holding for a short time. This indicates that based on the dilatometric curve of austenite transformation, the fraction of austenite formed at different temperatures could be accurately calculated through the use of the lever rule. Polynomial fitting was performed on the calculated results of austenite content ($f_{\gamma }$) in Figure 5b and obtained a mathematical model to predict the amount of austenite transformed during the austenitizing process, as shown in Equation (2).
(2)$$f_{\gamma }=-726112.7008\;+\;3686.0341T-7.0107T^{2}\;+\;0.0059T^{3}-1.8723\;\text{×}\;10^{-6}T^{4}$$
where *T* is the heating temperature, °C.

### 3.2. The Growth and Coarsening Process of Austenite

The growth and coarsening process of austenite during solution annealing at higher temperatures above A_c3_ have been explored. Figure 6 shows the microstructure of the samples after quenching from the single austenite zone. The microstructure of the samples was mainly martensite, and the martensitic lath became coarser with the increase in the quenching temperature, which indicated that austenite grains grew larger at a higher temperature.

The quenched samples were hotly etched with a saturated picric acid solution to analyze the original austenite grain, and the results are shown in Figure 7, Figure 8, Figure 9, Figure 10 and Figure 11. After etching, the original austenite grain boundary clearly appeared. Holding at 1100 °C, the austenite grains were relatively fine and uniform, and the austenite grain size changed little with the increase in the holding time. With the increase in the heating temperature, the austenite grain size increased, and the austenite grain grew obviously with the increase in the holding time. However, the growth rate of austenite grain was uneven, especially at higher temperatures. For example, the austenite grains had obvious size discrepancies when held at 1150 °C and 1200 °C for 120 min. After holding at 1250 °C for 30 min, austenite appeared with mixed grains, where the austenite grain size range was 25–190 μm. Additionally, as the holding time continued to extend, the small size of austenite grains gradually reduced. At 1300 °C, even when holding for 0 min, the austenite grain size distribution was uneven, as shown in Figure 11a. After holding for 30 min (see Figure 11b), some austenite grains became coarser, which resulted in seriously mixed grains, and the austenite grain boundary was relatively tortuous, indicating that the austenite growth process was not over. As the holding time increased, the austenite continued to grow through the Ostwald ripening mechanism. After holding for 120 min, the austenite grain boundary became straighter, where the angle at the triangular grain boundary was about 120°, and the austenite grain size was relatively uniform. At this time, due to the reduction in the grain boundary energy, the austenite grain grew slowly, as shown in Figure 11d.

The average grain size of austenite after solution treatment under different conditions was counted and the results are shown in Figure 12. The austenite grain size increased exponentially with the increase in temperature, and the longer the holding time, the faster the austenite grew with temperatures (see Figure 12a). While prolonging the holding time, the austenite grain size gradually increased. When the temperature was relatively low, such as 1100 °C and 1150 °C, the austenite grain size grew slowly with the holding time; however, with the increase in the temperature, the holding time had a significant effect on austenite grain size, such as for a temperature over 1200 °C, and the austenite grain began to grow rapidly when the holding time was over 60 min. The growth of austenite grain is a comprehensive physical metallurgical process integrating thermal activation, diffusion and an interfacial reaction. The main driving force is the total interfacial energy difference before and after grain growth, which is a spontaneous process that the whole system strives to achieve with the minimum total interfacial free energy [9,21]. This process was realized by the migration of atoms at the grain boundary. Therefore, at a low temperature, atomic diffusion was delayed, resulting in slow grain growth. In addition, the austenite grain growth needed not only driving force but also the mobility of the grain boundary. The precipitates in the sample could hinder the movement of the grain boundary, reducing its mobility, which also led to a smaller austenite grain size when the heating temperature was lower [12,22]. With the increase in the heating temperature, the driving force of austenite growth increased, and some precipitate began to dissolve into austenite, reducing the barrier of the grain boundary movement so that the austenite grain growth rate increased. Moreover, due to the uneven distribution of composition and precipitated particles in the sample, some of the austenite grains grew rapidly first with the increase in the holding time, and abnormal grain growth occurred, such as the mixed grains when held at 1250 °C for 30 min. Therefore, in this study, the growth rate of the grain size increased with the increase in the holding time when heating at higher temperatures. At 1300 °C, due to only a small number of precipitates existing, most austenite grains grew rapidly, and with the consumption of interfacial energy, the austenite grain growth rate decreased.

It can be seen that the growth of austenite grain with the increase in the holding temperature and time progressed through three stages, including the fine and uniform grain stage, the mixed grain stage with some abnormal grain growth and the coarse grain stage. Additionally, according to the evolution of austenite growth, the relationship between the temperature and time of the solution treatment could be divided into three regions, as shown in Figure 13. Holding at a low temperature, the austenite grain was fine and uniform. With the increase in the temperature, the holding time of entering the mixed grain zone and coarse grain zone decreased, and with a prolonged holding time, the mixed grains eventually grew into uniform coarse ones.

Figure 14 shows the results of the DSC experiment, where the main precipitate dissolution temperature was about 1133 °C under a heating rate of 10 °C/min. Based on the composition of the steel, the precipitates calculated by Thermo-Calc above 800 °C mainly consisted of Nb, Ti, C and N. Due to the bond strength among the Nb, Ti, N or C atoms being different, the stability of the precipitates formed by them was different at a high temperature. The dissolution temperature of Nb and Ti precipitates in the sample during heating could be calculated according to their solubility product in austenite, as shown in Formulas (3)–(6) [14,23]:(3)$$\log\left(\left[\mathrm{Ti}\right]\left[N\right]\right)=4.01-13850/T$$
(4)$$\log\left(\left[\mathrm{Ti}\right]\left[C\right]\right)=2.75-7500/T$$
(5)$$\log\left(\left[\mathrm{Nb}\right]\left[N\right]\right)=3.79-10150/T$$
(6)$$\log\left(\left[\mathrm{Nb}\right]\left[C\right]\right)=2.96-7510/T$$
where [Nb], [Ti], [C] and [N] are the concentration (wt%) of Nb, Ti, C and N, respectively. The average content of alloy elements into Formulas (3)–(6) and the calculated equilibrium precipitation temperature of each precipitate can be taken as *T*_TiN_ = 1424 °C, *T*_TiC_ = 1105 °C, *T*_NbN_ = 1037 °C and *T*_NbC_ = 1097 °C. It was found that except for the TiN particle, which had the highest dissolution temperature of 1424 °C, the dissolution temperature of the precipitates was around 1100 °C, which is basically consistent with the DSC results. That is, when the heating temperature was higher than 1150 °C, most of the precipitates dissolved into the austenite.

Figure 15 shows the precipitates in samples under different heat treatments analyzed by TEM. According to EDS analysis in Figure 15d,e, the samples mainly included two types of precipitates. One was the near-spherical Nb-rich (Nb, Ti)(C, N) precipitate, and the other one was the square Ti-rich (Nb, Ti)(C, N) precipitate. Because the precipitation temperature of TiN was the highest, it precipitated first during the cooling process of the casting billet. Due to the same crystal structure and close lattice constants of the carbides and nitrides of Nb and Ti, they had a coherent interface. Therefore, during the cooling process of the continuous casting slab, other precipitates, such as TiC and NbC, could easily nucleate and grow on TiN, forming composite precipitates [24,25]. Additionally, prior research has revealed that carbonitrides are more stable than pure carbides and need a higher temperature to fully dissolve into the austenite [26]. In steel, the content of the N element was relatively low, and the mass ratio of Ti/N was about 4.25, which is greater than the ideal ratio of 3.4, so N almost completely combined with Ti to form TiN, and the precipitates that formed in the lower temperature were mainly TiC and NbC. As shown in Figure 15a, at 1100 °C, the sample contained a large number of smaller spherical precipitates and larger square precipitates. The precipitates with an average size of ~18.6 nm had a strong effect on hindering the movement of austenite grain boundaries, so the austenite grains were relatively fine and uniform at this temperature. The relationship of the limit size *d* of the austenite grain, the volume fraction *f* and particle radius *r* of the precipitates are shown in Formula (7) [7].
(7)$$d=\frac{4}{3}\frac{r}{f(1+\cos\alpha )}$$
where *α* is the contact angle, which depends on the nature of the particle and has a weak effect. Formula (7) indicates that a smaller radius of precipitates with a large amount has a stronger inhibition effect on austenite growth. At 1200 °C, most of the precipitates were dissolved; the remaining particles were mainly square precipitates and a small number of spherical precipitates with an average size of ~34.3 nm. So, at this temperature, with the increase in the holding time, some austenite grains were rid of the pinning of precipitates and grew rapidly. At 1300 °C, the number of precipitates reduced greatly, and mainly square TiN precipitates remained with an average diameter of ~63.6 nm, which offered a small effect on inhibiting austenite grain boundary movement, so the austenite grain grew rapidly to a stable size at this temperature.

## 4. Conclusions

Based on Q960 steel, the evolution of austenite transformation and growth during heating treatment was systematically studied, and the main conclusions are as follows:The transformation of austenite is a nonlinear continuous process, and the amount of austenite gradually increased with the increase in the heating temperature and holding time, which could be accurately calculated by lever rule based on the dilatometric curve as the holding time within 10 min.With the increase in the holding temperature and time, the growth of austenite progressed through the fine and uniform grain stage, mixed grain stage with some grains with abnormal growth and coarse grain stage, and a heat treatment diagram, which was established to describe the evolution of austenite growth.The main precipitates in austenite are Nb-rich and Ti-rich (Nb, Ti)(C, N); the particle size increased, and the amount decreased with the increase in the heating temperature, which resulted in the rapid growth of austenite.

## Figures and Tables

**Figure 1 materials-16-03578-f001:**
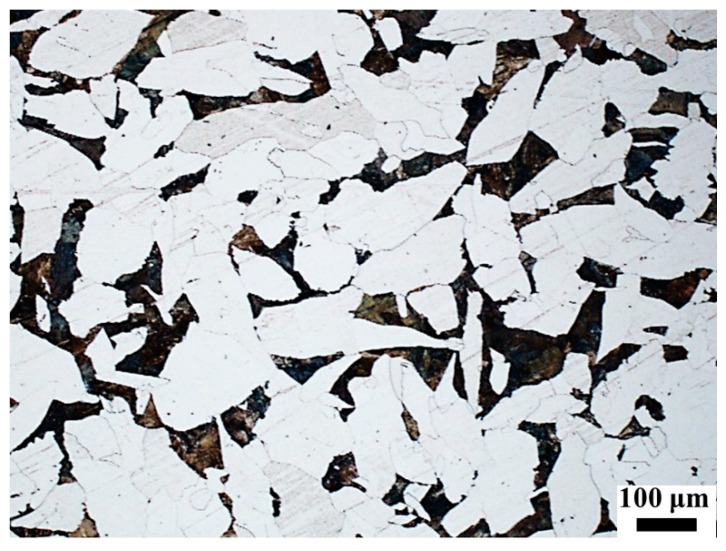
Microstructure of continuous casting slab at 1/4 thickness (the white phase is ferrite and the black phase is pearlite).

**Figure 2 materials-16-03578-f002:**
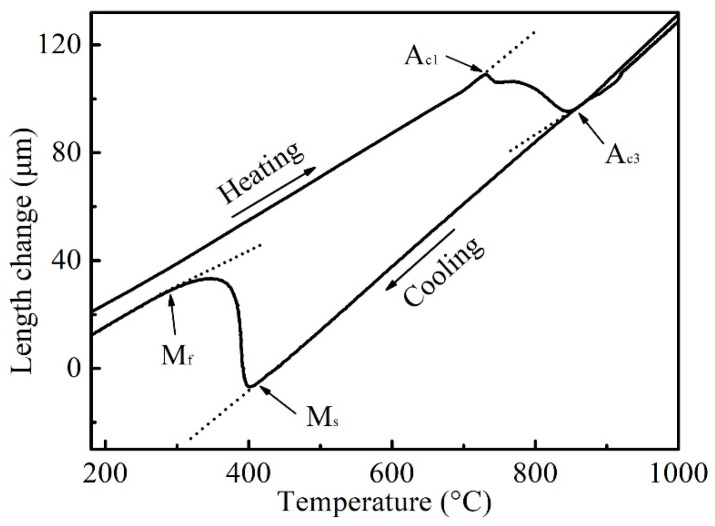
Dilatometric curve of sample length during heating and cooling process.

**Figure 3 materials-16-03578-f003:**
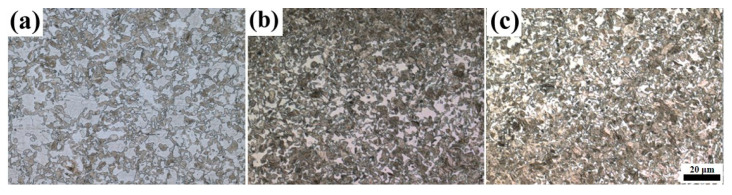
The microstructure of the samples cooled from different holding temperatures of (**a**) 790 °C × 10 min, (**b**) 790 °C × 30 min, (**c**) 790 °C × 60 min, (**d**) 810 °C × 10 min, (**e**) 810 °C × 30 min, (**f**) 810 °C × 60 min, (**g**) 830 °C × 10 min, (**h**) 830 °C × 30 min, (**i**) 830 °C × 60 min, (**j**) 850 °C × 10 min, (**k**) 850 °C × 30 min, (**l**) 850 °C × 60 min, (**m**) 880 °C × 10 min, (**n**) 880 °C × 30 min and (**o**) 880 °C × 60 min (the light phase is ferrite and the brown phase is martensite).

**Figure 4 materials-16-03578-f004:**
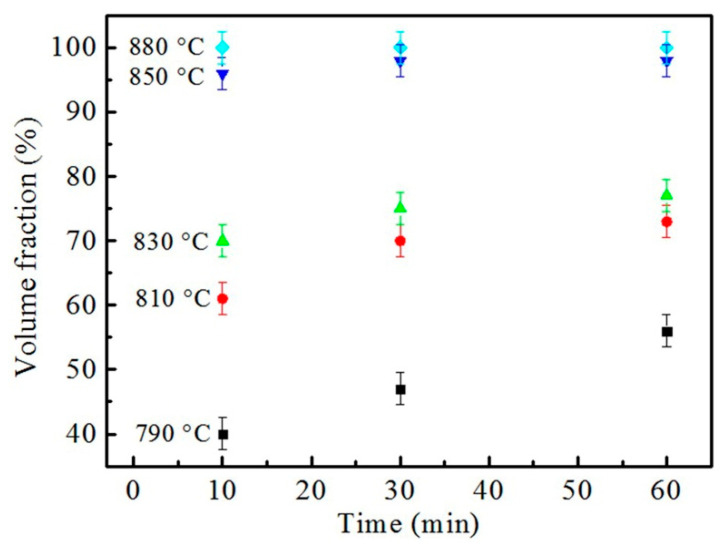
Variation in austenite content with holding temperature and time.

**Figure 5 materials-16-03578-f005:**
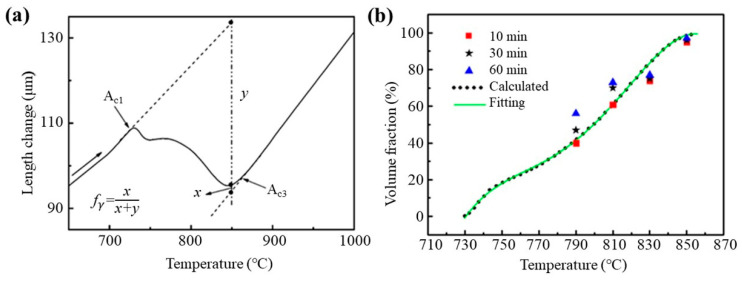
(**a**) Schematic diagram of lever rule of austenite transformation (the dashed straight lines are the extension of the straight dilatometric curve of the right part of austenite phase and the left part of mixed ferrite/pearlite phase), and (**b**) Relationship between austenite transformation and heating conditions.

**Figure 6 materials-16-03578-f006:**
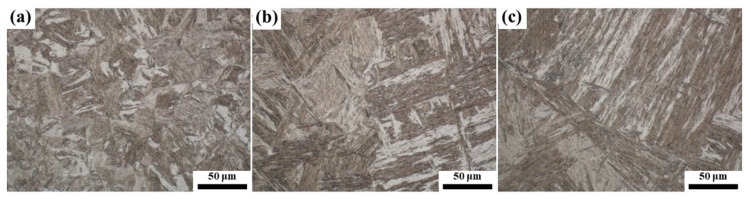
Microstructure of quenched samples after holding at (**a**) 1100 °C, (**b**) 1200 °C and (**c**) 1300 °C for 30 min.

**Figure 7 materials-16-03578-f007:**
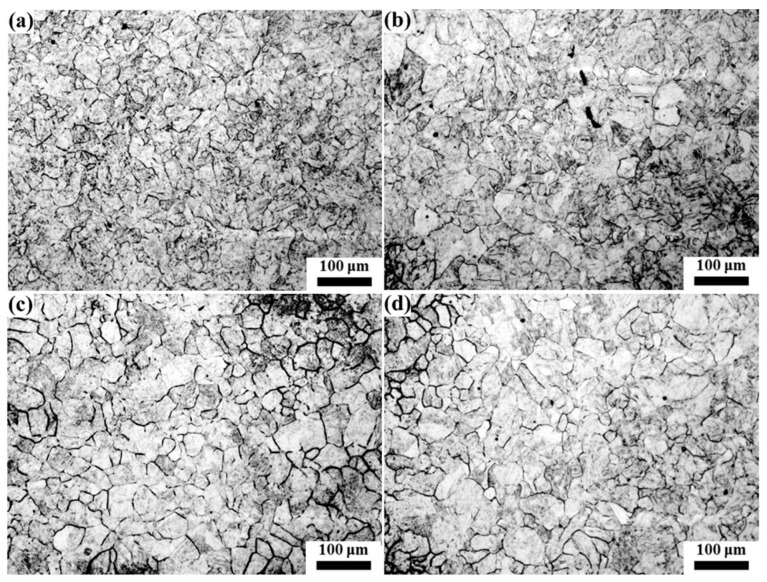
Morphology of original austenite grains at 1100 °C for (**a**) 0 min, (**b**) 30 min, (**c**) 60 min and (**d**) 120 min.

**Figure 8 materials-16-03578-f008:**
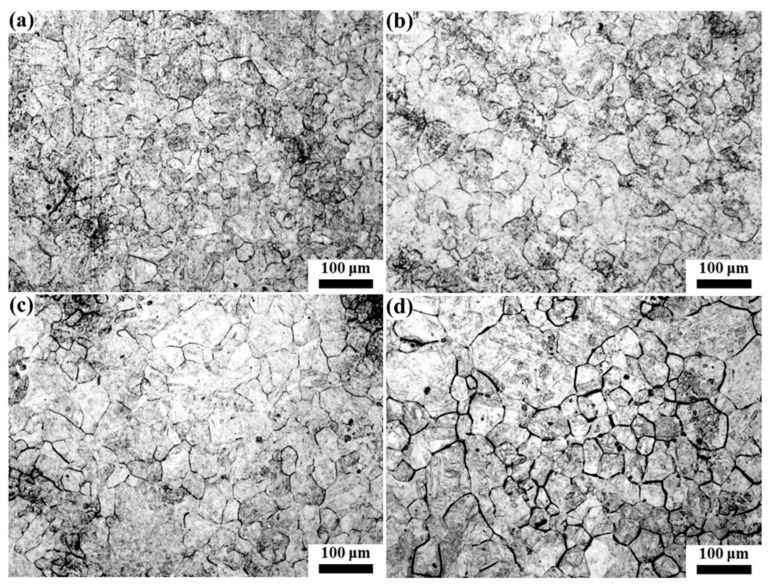
Morphology of original austenite grains at 1150 °C for (**a**) 0 min, (**b**) 30 min, (**c**) 60 min and (**d**) 120 min.

**Figure 9 materials-16-03578-f009:**
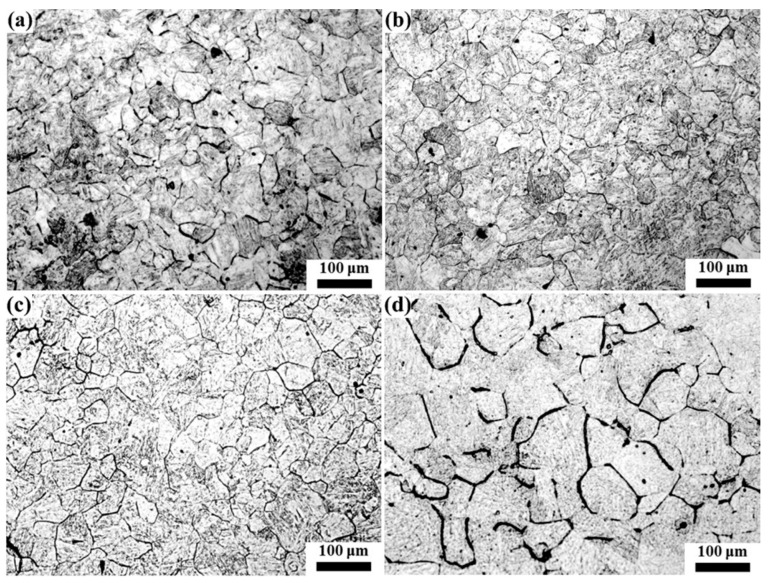
Morphology of original austenite grains at 1200 °C for (**a**) 0 min, (**b**) 30 min, (**c**) 60 min and (**d**) 120 min.

**Figure 10 materials-16-03578-f010:**
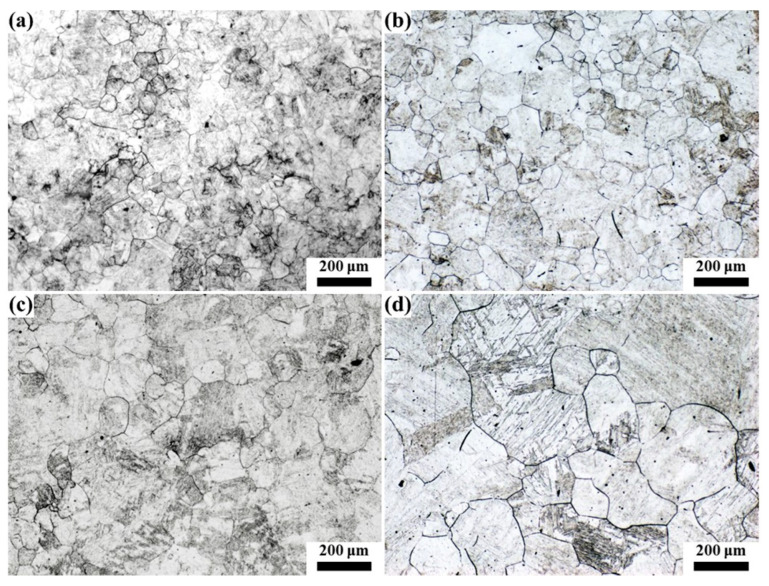
Morphology of original austenite grains at 1250 °C for (**a**) 0 min, (**b**) 30 min, (**c**) 60 min and (**d**) 120 min.

**Figure 11 materials-16-03578-f011:**
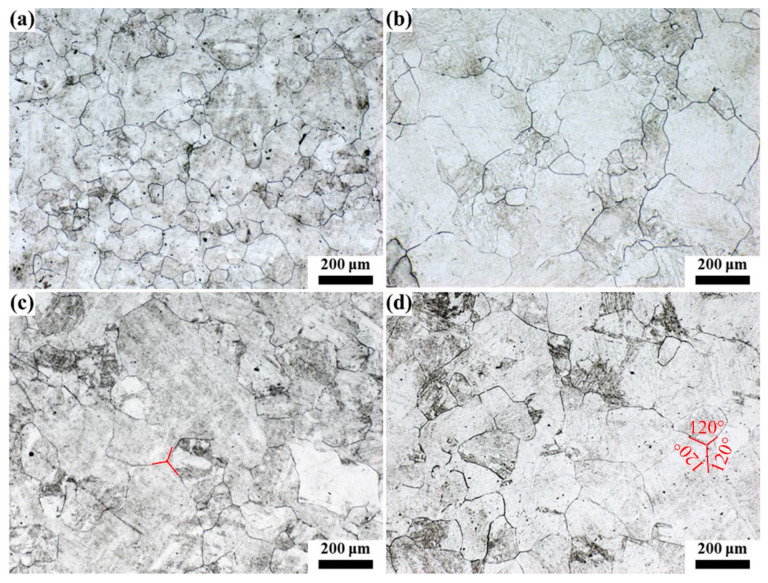
Morphology of original austenite grains at 1300 °C for (**a**) 0 min, (**b**) 30 min, (**c**) 60 min and (**d**) 120 min.

**Figure 12 materials-16-03578-f012:**
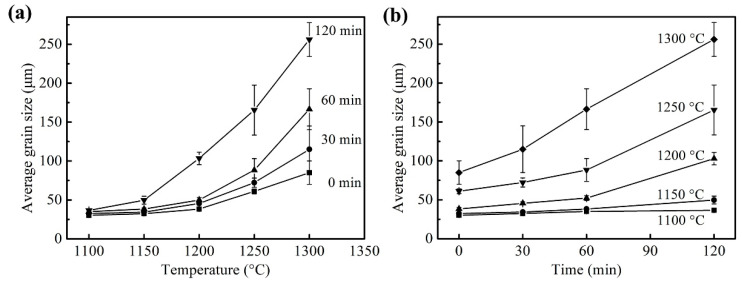
Variation in austenite grain size with (**a**) Heating temperature and (**b**) Holding time.

**Figure 13 materials-16-03578-f013:**
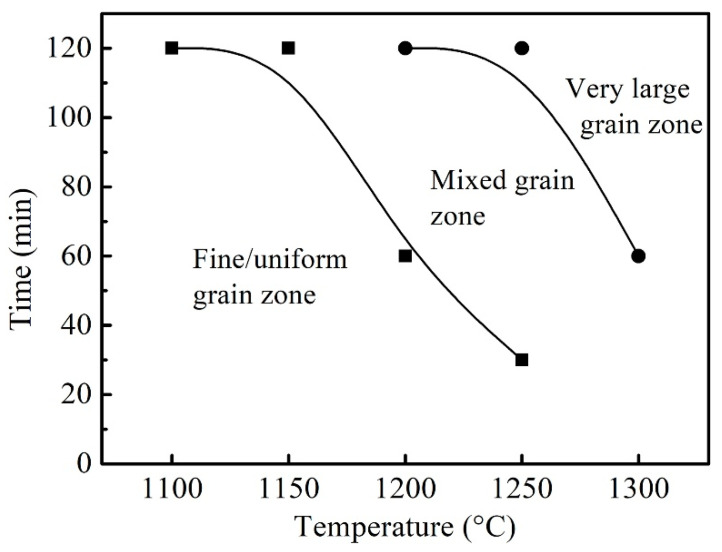
Temperature—time relationship diagram of austenite grain growth.

**Figure 14 materials-16-03578-f014:**
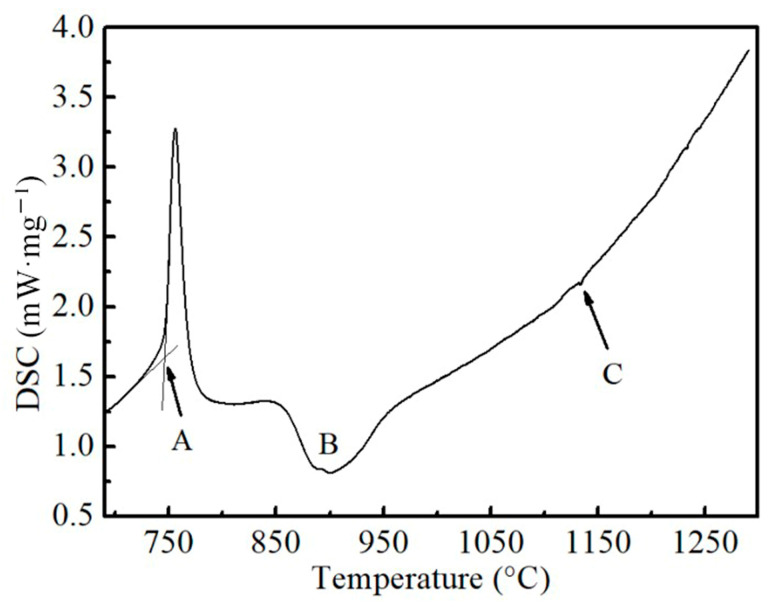
DSC curve of the sample during heating process (A represents A_c1_, B represents A_c3_, and C represents precipitate dissolution temperature).

**Figure 15 materials-16-03578-f015:**
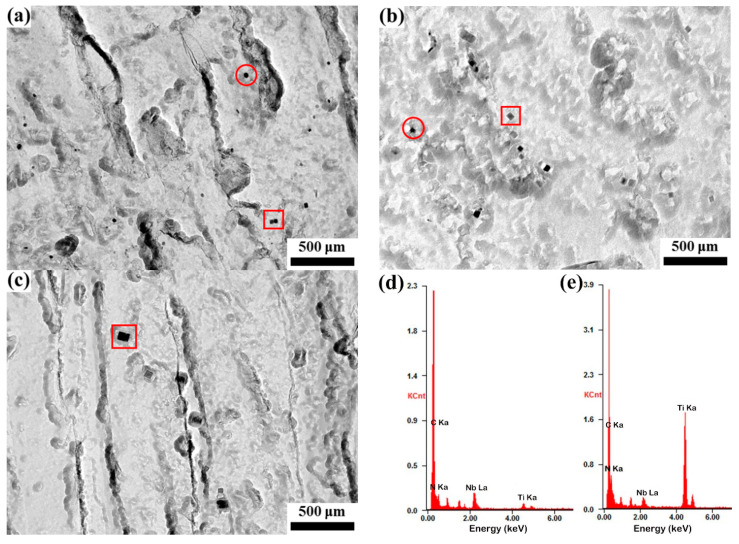
The morphology of precipitates in the samples after holding at (**a**) 1100 °C, (**b**) 1200 °C and (**c**) 1300 °C for 30 min, and EDS spectrum of (**d**) spherical precipitates in red circular and (**e**) cubic precipitates in red square.

**Table 1 materials-16-03578-t001:** Chemical composition of the investigated steel (wt%).

C	Si	Mn	Cr	Mo	Nb	Ti	N	P	S	Fe
0.12	0.30	1.30	0.25	0.10	0.02	0.017	<0.004	<0.01	<0.003	Bal.

## Data Availability

The data presented in this study are available upon request from the corresponding author.

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
