# Peer review of "Study on Austenite Transformation and Growth Evolution of HSLA Steel"

_materials, 2023, doi:10.3390/ma16093578_

Round 1

Reviewer 1 Report

The authors investigated the influence of temperature and time on the transformation and growth of austenite. They present good and sufficient results, and some points should be revised:

- Line 134 (Figure 2): “… and then ferrite begins to transform into austenite at higher temperature (such as 830°)”. The temperature of ferrite to austenite transformation can be determined according to Ref. 19.

- Equation 1 and Figure 5 (lever rule) should be revised.

- Figure 12: Include the deviations in the figure.

- Figure 14: Is C the only peak of precipitate dissolution? 

- Line 76: Error Table 1;

- Equation 3 to 6: Change lg for log.

Author Response

The authors investigated the influence of temperature and time on the transformation and growth of austenite. They present good and sufficient results, and some points should be revised:

Point 1: - Line 134 (Figure 2): “… and then ferrite begins to transform into austenite at higher temperature (such as 830 ℃)”. The temperature of ferrite to austenite transformation can be determined according to Ref. 19.

Response 1: Thank you for your good comments. According to Ref. 19 we get the start transformation temperatures of 730 ℃ and 745 ℃ for pearlite and ferrite transforming to austenite, respectively. And we have revised the sentence in Line 134 into “where the pearlite first transforms into austenite at low temperature (such as 730 ℃), and then ferrite begins to transform into austenite at higher temperature (such as 745 ℃) [19]” in the revised manuscript.

Point 2: - Equation 1 and Figure 5 (lever rule) should be revised.

Response 2: Thank you for your valuable comments. We have corrected the equation in Figure 5 (a) to  in the revised manuscript.

Point 3: - Figure 12: Include the deviations in the figure.

Response 3: Thank you for your valuable comments. We have added the deviations in the Figure 12 of the revised manuscript.

Point 4: - Figure 14: Is C the only peak of precipitate dissolution?

Response 4: Thank you for your valuable comments. Due to the dissolution temperature of precipitates is different, there must be other peaks in the DSC curve, which are not obvious. In this work C is the obvious peak of precipitate dissolution on the DSC curve, which means that most precipitates, such as (Ni, Ti)(C, N), dissolve into austenite during heating process.

Point 5: Comments on the Quality of English Language

- Line 76: Error Table 1;

- Equation 3 to 6: Change lg for log.

Response 5: The “Table” in Line 76 has been corrected to “table” and “Figure” in Line 114, 123, 131,136, 168, 169, 184, 195, 216, 226, 227, 234, 251, 253, 284, 311 and 323 has been corrected to “figure”. And we have changed lg in equation 3 to 6 to log.

Reviewer 2 Report

Authors of the publication "Study on austenite transformation and growth evolution of HSLA steel" presented the systematic study of the austenite transformation and growth behavior during the heating process based on a continuous casting slab of Q960 steel. The transformation process of austenite at the intercritical temperature was characterized with a thermal expansion instrument, and by combining microstructure analysis, a mathematical model was developed to predict the austenite transformation. The growth behavior of austenite at temperatures above Ac3 was discussed in detail, and the effect of precipitates on austenite growth was explored based on TEM and DSC experiments.

The authors' research provides valuable guidance on heat treatment for obtaining an optimal microstructure and mechanical properties in HSLA steels.

The first part of the publication was devoted to a literature review of the subject selected by the authors. This part has been described very carefully and in detail. The next part concerns the materials and methods and has also been discussed in great detail. Each of the graphs and figures is presented in a very legible and clear way, and their interpretation is described in detail. The conclusions are consistent and closely related to the research topic.

As a reviewer of this work, however, I believe that the reviewed work requires corrections of stylistic errors. In publications of this type for the journal "Materials", such errors are unacceptable, so they should be corrected. For example:

1. Line 81 - "and then rapidly cooled at 3000 /min to room temperature" - This seems to be an error, as 3000 /min is an extremely high cooling rate.

2. Patterns 3-6 - "correct lg to log" - This is a minor correction to the labeling of the patterns.

3. Line 229 - "where the angle at triangular grain boundary is about 120°" - Please mark the example angle on the figure. - This is a clarification that could be helpful for readers to better understand the findings.

In summary, this publication addresses important issues related to heat treatment.

Author Response

Response to Reviewer 2 Comments

Authors of the publication "Study on austenite transformation and growth evolution of HSLA steel" presented the systematic study of the austenite transformation and growth behavior during the heating process based on a continuous casting slab of Q960 steel. The transformation process of austenite at the intercritical temperature was characterized with a thermal expansion instrument, and by combining microstructure analysis, a mathematical model was developed to predict the austenite transformation. The growth behavior of austenite at temperatures above Ac3 was discussed in detail, and the effect of precipitates on austenite growth was explored based on TEM and DSC experiments.

The authors' research provides valuable guidance on heat treatment for obtaining an optimal microstructure and mechanical properties in HSLA steels.

The first part of the publication was devoted to a literature review of the subject selected by the authors. This part has been described very carefully and in detail. The next part concerns the materials and methods and has also been discussed in great detail. Each of the graphs and figures is presented in a very legible and clear way, and their interpretation is described in detail. The conclusions are consistent and closely related to the research topic.

As a reviewer of this work, however, I believe that the reviewed work requires corrections of stylistic errors. In publications of this type for the journal "Materials", such errors are unacceptable, so they should be corrected. For example:

Point 1: Line 81 - "and then rapidly cooled at 3000 ℃/min to room temperature" - This seems to be an error, as 3000 ℃/min is an extremely high cooling rate.

Response 1: Thank you for your good comments. The samples are cooled through liquid nitrogen in the German Beahr DIL 805A thermal expansion instrument, and the cooling rate can exceed 50 ℃/s for small samples, so it can cool the sample with size of Φ 4 mm × 10 mm at 3000 ℃/min here.

Point 2: Patterns 3-6 - "correct lg to log" - This is a minor correction to the labeling of the patterns.

Response 2: Thank you for your valuable comments. The patterns of “lg” in equation 3 to 6 have been corrected to “log” in the revised manuscript.

Point 3: Line 229 - "where the angle at triangular grain boundary is about 120°" - Please mark the example angle on the figure. - This is a clarification that could be helpful for readers to better understand the findings.

Response 3: Thank you for your valuable comments. Example angles have been marked on the figure 11 c and d in the revised manuscript.

Reviewer 3 Report

Paper is written with sound Technical Contents.

Author Response

Response to Reviewer 3 Comments

Point 1 :Paper is written with sound Technical Contents.

Thanks a lot for your careful review of the manuscript and offering us valuable comments and suggestions.

Round 2

Reviewer 2 Report

In the article presented for review, the authors described their research very meticulously and with great care. All of the results contained in it were presented in a very clear and legible way for the recipient. The conclusions and the study summary are consistent and well-formulated. Summing up, the reviewed work presents a very high substantive and experimental value.